# Secondary Breast Malignancy from Renal Cell Carcinoma: Challenges in Diagnosis and Treatment—Case Report

**DOI:** 10.3390/diagnostics13050991

**Published:** 2023-03-05

**Authors:** Marko Spasic, Dusan Zaric, Minja Mitrovic, Sanja Milojevic, Nikola Nedovic, Marija Sekulic, Bojan Stojanovic, Dejan Vulovic, Bojan Milosevic, Filip Milutinovic, Neda Milosavljevic

**Affiliations:** 1Department of Surgery, Faculty of Medical Sciences, University of Kragujevac, 34000 Kragujevac, Serbia; 2Clinic for General Surgery, University Clinical Centre, 34000 Kragujevac, Serbia; 3Clinic for Urology, Clinical Hospital Centre “Dragisa Misovic”, 11000 Belgrade, Serbia; 4Faculty of Medical Sciences, University of Kragujevac, 34000 Kragujevac, Serbia; 5Centre for Radiology, University Clinical Centre, 34000 Kragujevac, Serbia; 6Department of Hygiene and Ecology, Faculty of Medical Sciences, University of Kragujevac, 34000 Kragujevac, Serbia; 7Centre for Plastic Surgery, University Clinical Centre, 34000 Kragujevac, Serbia; 8Clinic for Urology, University Clinical Centre, 34000 Kragujevac, Serbia; 9Centre for Radiation Oncology, University Clinical Centre, 34000 Kragujevac, Serbia

**Keywords:** secondary malignancy, breast, renal cell carcinoma, metastasis, metastasectomy

## Abstract

Renal cell carcinoma represents about 2% of all malignant tumours in adults. Metastases of the primary tumour in the breast make up to about 0.5–2% of the cases. Renal cell carcinoma metastases in the breast are extremely rare and have been sporadically recorded in the literature. In this paper, we present the case of a patient with breast metastasis of renal cell carcinoma 11 years after primary treatment. Case presentation: An 82-year-old female who had right nephrectomy due to renal cancer in 2010 felt a lump in her right breast in August 2021, whereby a clinical examination revealed a tumour at the junction of the upper quadrants of her right breast, about 2 cm, movable toward the base, vaguely limited, and with a rough surface. The axillae were without palpable lymph nodes. Mammography showed a circular and relatively clearly contoured lesion in the right breast. Ultrasound showed an oval lobulated lesion of 19 × 18 mm at the upper quadrants, with strong vascularisation and without posterior acoustic phenomena. A core needle biopsy was performed, and the histopathological findings and obtained immunophenotype indicated a metastatic clear cell carcinoma of renal origin. A metastasectomy was performed. Histopathologically, the tumour was without desmoplastic stroma, comprising predominantly solid-type alveolar arrangements of large moderately polymorphic cells, bright and abundant cytoplasm, and round vesicular cores with focally prominent nuclei. Immunohistochemically, tumour cells were diffusely positive for CD10, EMA, and vimentin, and negative for CK7, TTF-1, renal cell antigen, and E-cadherin. With a normal postoperative course, the patient was discharged on the third postoperative day. After 17 months, there were no new signs of the underlying disease spreading at regular follow-ups. Conclusion: Metastatic involvement of the breast is relatively rare and should be suspected in patients with a prior history of other cancers. Core needle biopsy and pathohistological analysis are required for the diagnosis of breast tumours.

## 1. Introduction

Kidney cancer accounts for approximately 5% of cancer incidence worldwide, with higher incidence and mortality rates in men. This difference can be explained by genetics, sex hormones, and modifiable risk factors such as uncontrolled hypertension, alcohol, and obesity [1]. The most common type of kidney cancer in the cortex is renal cell carcinoma (RCC), accounting for approximately 2% of adult malignancies [2] and encompassing 80–85% of all renal neoplasms, predominantly the clear cell RCC subtype (75–85%). Other RCC subtypes include papillary, chromophobe, oncocytoma, and other [3]. Renal pelvis cancer characteristics resemble those of urothelial cancer [2]. In most cases, RCC is diagnosed incidentally, and about 10% of patients experience the classic triad: palpable mass, flank pain, and haematuria. Other common symptoms include weight loss, fatigue, fever, and others, with 20% of patients diagnosed with a variety of paraneoplastic syndromes [2]. Renal cell cancer differs in biology and behaviour, reflecting multiple genetic foundations and well-defined hereditary renal cancer syndromes. The practical significance of this in terms of treatment is limited, while certain renal cell neoplasms are subject to specific therapy recommendations [4]. The most common metastasis in renal cell carcinoma occurs in the lungs, followed by the involvement of the bones, liver, lymph nodes, adrenal glands, and brain [5]. About one-third of patients with renal cell carcinoma have metastases at the time of diagnosis or within months or years after renal cell cancer treatment [2]. For patients with localised disease, the recommended treatment is surgery, with no benefit from adjuvant therapy. In advanced settings, a combination of tyrosine kinase inhibitor and PD-1/PD-L1 inhibitor shows benefits in terms of increasing response rates and prolonging progression-free survival [6]. When considering treatment options, the recommendations by the International Metastatic Renal Cell Carcinoma Database Consortium (IMDC), which stratifies patients with metastatic disease into three subgroups—good-, intermediate-, and poor-risk groups, should be reviewed [7]. Breast cancer is the most common cancer in the world, with a 47.8% incidence rate and 2,261,419 cases in 2020, according to the Global Cancer Observatory (GLOBOCAN) [8]. Breast metastases from extramammary tumours are rare, with an incidence ranging from 0.5% to 2% in clinical reports [9] on all breast neoplasms; in most cases, they are accompanied by a poor prognosis. Diagnosed breast masses in up to 50% of cases may be the first sign of another primary tumour [10]. It is recommended that treatment and its sequencing follow the guidelines for the primary tumour type [10]. Metastases in the breast from renal cell carcinoma are extremely rare, with only a few sporadic cases reported in the literature [11]. In this study, we report a rare case involving a female patient who underwent a nephrectomy and, 11 years later, was diagnosed with a breast metastasis from the renal cell carcinoma. Based on the literature research we performed, this is the first case from Serbia to be published.

## 2. Case Presentation

Data from the patient’s complete medical record, which included clinical and radiological findings from the treatment of RCC, as well as clinical breast examinations, breast ultrasound, mammography, CT findings of the chest, abdomen, and pelvis, and histopathological findings, were used for writing this case report. The patient gave her written consent to publish the data. An 82-year-old female patient underwent complete diagnostics due to the appearance of haematuria, and it was determined that she had a primary tumour of the right kidney. No distant metastases on other organs were observed. The oncology MDT suggested a radical right nephrectomy. According to the American Joint Committee on Cancer (AJCC), 7th edition, the disease was Stage I, T1bN0M0. She had regular urological follow ups until August 2021, when she felt the lump in her right breast and consulted a doctor. Clinical breast examinations detected a 2 cm movable toward the base, vaguely limited, rough-surfaced lump, without clear-cut borders at the junction of the upper quadrants of the right breast. No other palpable tumour lesions were observed in the rest of the right breast or in the left breast. The axillary lymph nodes were not detected. Mammography was performed and showed a mass with well-defined borders in the right breast at the 12-1 o’clock position, 4 cm from the nipple (Figure 1).

Suspicious breast calcifications were not detected, and the axillae were without significant lymph nodes. Skin and subcutaneous tissue were without signs of infiltration. Ultrasound of the right breast was performed, showing an oval lobulated lesion of 19 × 18 mm at the upper quadrants with strong vascularisation, with no posterior acoustic phenomena (Figure 2A). A 14-gauge core needle biopsy was performed under local anaesthesia, and three samples of the tumour mass were obtained. The examined material contained a section of breast tissue, with fibroadipose tissue permeated with tubular pseudoglandular arrangements, round and polygonal cells, distinctly bright cytoplasm, and uniform centrally placed nuclei with a pronounced capillary network between them. Immunohistochemically, tumour cells express EMA, AE1/3, CD10, and vimentin, and a moderate amount of CK7, but do not express ER, PR, or HER2. Proliferation index Ki-67 expression was 10%. Histopathologically and immunohistochemically, the tumour was metastatic RCC. A multislice CT scan of the chest, abdomen, and pelvis w performed when staging the cancer diagnosis. This showed a hypervascular tumour 2.2 × 2.4 cm, whereas the axillae were without significant lymph nodes (Figure 2B).

There were no signs of right kidney cancer recurrence, and the other findings of the chest, abdomen, and pelvis showed no signs of disease. The patient was presented to a urology and breast multidisciplinary team, and a metastasectomy was suggested. A partial right breast resection and a histological and immunohistochemical analysis of the sample were performed (Figure 3).

Histopathologically, the tumour was without desmoplastic stroma and mainly comprised solid-type alveolar arrangements of large moderately polymorphic cells, bright and abundant cytoplasm, and round vesicular cores with focally prominent nuclei (Figure 4). Immunohistochemically, the tumour cells were diffusely positive for CD10, EMA (epithelial membrane antigen), and vimentin, and negative for CK7, TTF-1, renal cell antigen, and E-cadherin with positive and negative, and internal and external controls for the given antibodies (Figure 4).

The aforementioned was a histological confirmation of metastatic renal cell carcinoma. Due to a normal postoperative course, the patient was discharged on the third postoperative day. The postoperative multidisciplinary team suggested regular follow-up appointments. After 17 months, no new signs of underlying disease spread were observed. The patient felt well, with a good (ECOG = 1) performance status.

## 3. Discussion

Breast metastases from extramammary primary malignancies are rare where both breasts are equally involved, with uncommon bilateral involvement. In about 85% of cases, solitary lesions are found in the breast [12]. Breast metastases, although rare, mainly originate from melanoma, lymphoma, and leukaemia, as well as the ovaries, lungs, and stomach [13]. Disease manifestation can vary, showing signs and symptoms of primary neoplasms and/or metastatic breast lesions, and is influenced by dissemination [10]. Clinical practice and the literature show that renal cell carcinoma can metastasize to any part of the body, but unusual metastatic sites, as described in published case reports, are the ovaries, pancreas, spleen, and head and neck organs, such as the thyroid [14]. Routine surveillance imaging does not include the aforementioned organs, which delays the identification of metastatic disease, resulting in patients being diagnosed after exhibiting symptoms [14]. The recurrence risk for RCC is highest after treatment, with median relapse time within the first two years, but many case reports, especially those with unusual metastatic sites, have reported disease recurrence after more than a decade after initial surgical treatment, where surveillance imaging does not provide for recommendations [14], as in our case, with recurrence 11 years after nephrectomy. Similar data are shown in other published cases that report RCC metastasis in the breast. The route for metastatic tumour cells includes the right ventricle of heart, then through the inferior vena cava, passing into the pulmonary circulation, and eventually the breast. The paravertebral venous plexus can be involved in cancer cell transport as well [15]. Regardless of the primary malignancy source, the presence of breast metastasis is associated with a poor outcome, with a mean survival of 10.9 months [16]. The available published case reports (PubMed database) show that all patients reported in the cases were women. Clinically, breast metastatic lesions, unlike primary tumours, do not involve the skin, nor do they cause nipple retraction or discharge [12], whereas the involvement of an axillary lymph node is variable [16]. Primary breast malignancies in most of the cases have spiculated lesions, microcalcifications, architectural distortion, and asymmetrical density [17], while radiographic findings of secondary tumours showed features that are not metastases-specific—nonspiculated lesions, oval-shaped, without calcifications, and well-circumscribed, similar to most benign tumours [11,18], except when they have rich blood flow, as is in this case, where lesions show increased vascularity. In most of the cases, clear cell RCC is hypervascular [19], due to a defective von Hippel Lindau (VHL) tumour suppressor gene causing the upregulation the vascular endothelial growth factor (VEGF) and platelet-derived growth factor (PDGF), which promote survival, tumour cell growth, and angiogenesis [20]. Pathology analysis of our case showed metastatic RCC, initially treated with radical nephrectomy 11 years prior to the current presentation, whereas immunohistochemistry revealed positive staining for CD10, EMA, and vimentin, which was consistent with the diagnosis of RCC [21], and negative for CK7, TTF-1, renal cell antigen, and E-cadherin. Metastatic disease in RCC, regardless of the site, can occur in synchronous form—at the time of the primary cancer diagnosis, or metachronous form (as in the presented case), when metastases are diagnosed in a later follow up [22]. In the Republic of Serbia, according to real-world data, there were 1155 newly diagnosed patients per year (in 2020), with a 5-year prevalence of 35.56 per 100,000 people with RCC [8]. However, this is first recorded case of isolated, metachronous RCC in breast tissue in Serbia. Patients with synchronous metastasis have a worse cancer-specific survival rate compared with that for the metachronous pattern of the disease [22,23]. Early suspicion and disease recurrence confirmation are fundamental for future therapeutic management. The standard diagnostic battery of tests includes, as described in our case, breast imaging and radiographic body evaluation to determine whether the disease is oligoprogressive. Over the past years, due to an increase in patients with breast masses requiring precise and individual diagnostics testing, the necessity for advanced diagnostics has arisen.

Breast tomosynthesis overcomes the limitations of mammography due to the ability to provide three-dimensional information (using a lower dose), enhancing the early detection of breast neoplasms. Dedicated breast computed tomography (DBCT) is a technique comparable to magnetic resonance imaging (MRI); however, it is possible to perform it without breast compression, and it is not limited by breast density or implants. Breast MRI is recommended in diagnosis, staging, and at follow up, using diffusion-weighted imaging (DWI). Utilising apparent diffusion coefficient (ADC) values, this shows a promising ability to characterise breast lesions [24]. The use of artificial intelligence (AI), such as algorithms and computational models, may represent a future answer to challenges in precise breast lesion diagnosis [24] and facilitate treatment decisions. Due to the rare occurrence of this case presentation, no accepted consensus or specific management treatment guidelines exist. Metastatic breast RCC is treated following principles that are used to treat other metastatic lesions [11], whereby surgical resection may be a recommended management option for patients with isolated metastases, especially in those with a prolonged metachronous disease-free interval [6]. In less than 15% of cases, a complete response to nonsurgical treatment was achieved, while surgical treatment is associated with better overall survival [25]. Patients with metastatic RCC have a poor prognosis, and their overall survival differs in the literature, but without considering treatment modalities, patients had a median survival of 5.9 months (for the high-risk group) to 50.6 months (for the low-risk group) [26], while our patient continued to undergo regular follow-up examinations and showed no evidence of disease 17 months after the diagnosis of the metastatic disease. There has been a rapid expansion of therapeutic options for these patients in the last few years, in particular targeted immunotherapy, with CR rates suggesting that it may be possible to achieve a survival benefit [27]. Furthermore, stereotactic ablative radiotherapy is safe and efficacious for RCC oligometastases, with a high one-year local control (at 90%) and can be recommended as an option in selected cases [28], either alone or in combination with systemic treatment [29].

## 4. Conclusions

Metastases of primary tumours in the breast are extremely rare. Due to the rare occurrence of this case presentation, there are no studies with larger patient numbers and randomised therapeutic options or any accepted consensus or specific treatment guidelines, despite rapidly growing therapeutic options for these patients over the last decade. It is necessary to consider this rare localisation of metastases, especially in patients with a history of treated malignancies of other primary localisations. To establish the correct diagnosis of a metastatic tumour in the breast, core needle biopsy, pathohistological analysis, and immunohistochemical analysis are mandatory because they indicate the type and origin of the metastatic tumour in the breast so that adequate modalities of systemic or surgical treatment can be applied.

## Figures and Tables

**Figure 1 diagnostics-13-00991-f001:**
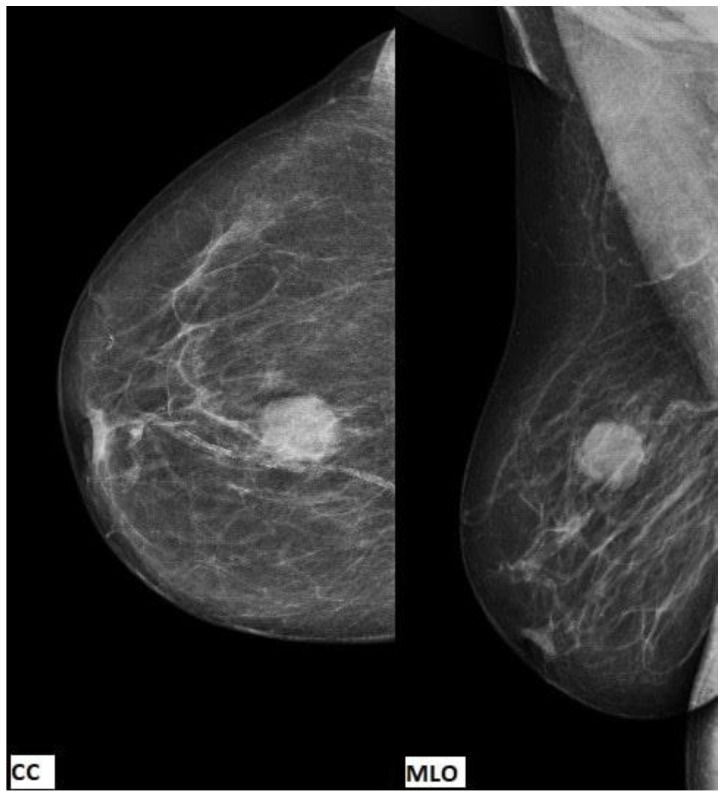
The cranial caudal (CC) view and the mediolateral oblique (MLO) view showing a mass with well-defined borders in the right breast, without suspicious breast calcifications or axillar lymph nodes.

**Figure 2 diagnostics-13-00991-f002:**
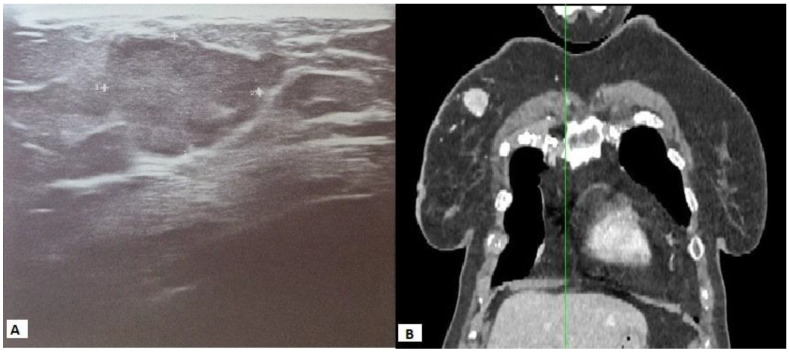
Ultrasound of the right breast showing an oval lobulated lesion of 19 × 18 mm with strong vascularisation, without posterior acoustic phenomena (**A**); multislice CT scan of the chest showing the right breast’s hypervascular tumour of 2.2 × 2.4 cm, whereas the axillae were without significant lymph nodes (**B**).

**Figure 3 diagnostics-13-00991-f003:**
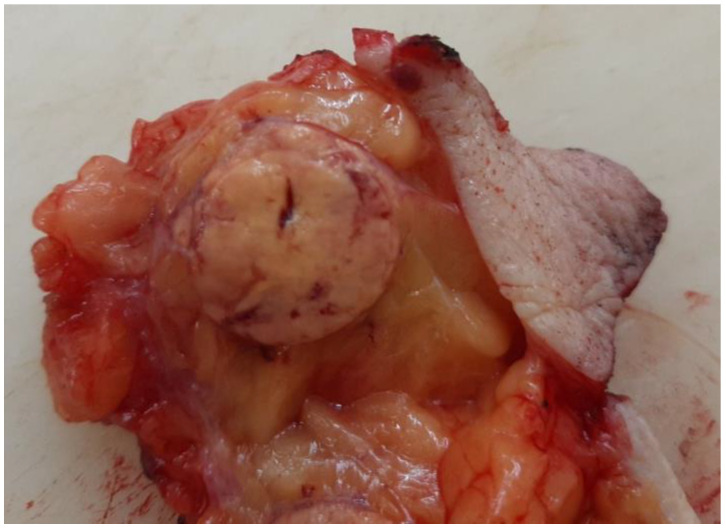
Macroscopic findings showing the tumour in the right breast with clear boundaries, yellow brown in colour, and 18 mm in diameter.

**Figure 4 diagnostics-13-00991-f004:**
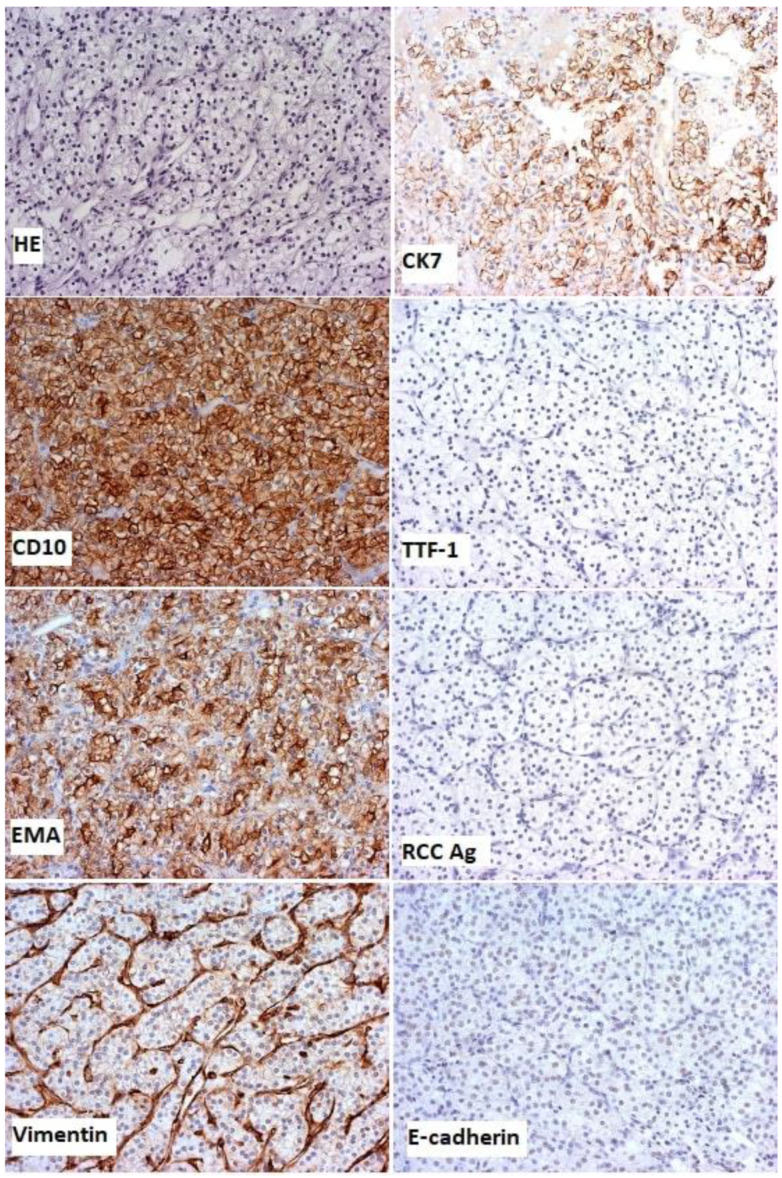
Histological examination, haematoxylin and eosin staining (HE), and immunohistochemical staining with magnification ×200.

## Data Availability

Not applicable.

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
