# Peer review of "Secondary Breast Malignancy from Renal Cell Carcinoma: Challenges in Diagnosis and Treatment—Case Report"

_diagnostics, 2023, doi:10.3390/diagnostics13050991_

Round 1

Reviewer 1 Report

Grammar and spelling of English language need to minor revision.

Author Response

According to Assistant Editor suggestion we extended our manuscript to at least 2500 words.

Reviewer 2 Report

  Thank you for requesting  to provide a review of this article, which has a subject of high interest. 

   The main purpose of the analysis was to present the case of a patient with breast metastasis of renal cell carcinoma (RCC), 11 years later after primary treatment.

   The main question adressed in the research was whether renal cell cancer may undergo meastasis in the breast, as it is extremely rare for this type of cancer and, therefore, the study reports a case of a 82 years old female patient, who underwent nephrectomy for a renal cell carcinoma and who presented 11 years later with a breast metastasis from RCC. Apparently, this was the first case to be published from Serbia. 

   The study is a case report analysis, following an 82-year old women who presented for a breast  metastasis after 11 years from the nephrectomy. The topic is original and relevant in the field and brings usefull knowledge regarding the subject. A comprehensive search strategy was used. The review methodology was comprehensive with screening and data extraction. When it comes to the methodology used, no specific improvements should be considered from my point of view.

   The conclusions are consistent with the evidence and the arguments presented, and they adress properly to the main question which conducted the analysis.

   The references are appropriate and well suited for this kind of study. 

    Regarding the figures and pictures used in the article, they provide suitable information about the cases and show significant statistical references. They are also well understandable and the information is easy to be followed. There are no other comments required about these items, from my point of view.

  Regarding the structure and accuracy of the phrases, the manuscript has well structured information, with supported evidence and well structured phrases.

   The manuscript is original and well defined. The results provide an advance in current knowledge. The results are being interpreted appropriately and are significant, as well as the conclusions.

  The article is written in an appropriate way. 

  The study is correctly designed and the analysis is being performed at high standards, so the data are robust enough to draw the conclusion. 

   Surely the paper will attract a wide readership. 

   The English language is appropriate and well understandable.

   There are a few things to add in the lines below, but the article should be published after the corrections are made: 

Line 20: make up to about, not „make up about”

Line 27: described, not „describe”

Line 28: showed, not „shows”

Line 30: indicated, not „indicate”

Line 31: was, not „is”

Line 34: were, not „are”

Line 72: was, not „is”

Line 92: was, not „is”

Line 95: were, not „are”

Author Response

(The authors gave the same response as above.)
